# Compressive Failure Mechanism of Structural Bamboo Scrimber

**DOI:** 10.3390/polym13234223

**Published:** 2021-12-02

**Authors:** Xueyu Wang, Yong Zhong, Xiangya Luo, Haiqing Ren

**Affiliations:** Research Institute of Wood Industry, Chinese Academy of Forestry, Beijing 100091, China; wangxueyu1784@163.com (X.W.); zhongyong108@163.com (Y.Z.); luoxiangya0423@sina.cn (X.L.)

**Keywords:** compression performance, load–displacement curves, digital image correlation (DIC), failure mode

## Abstract

Bamboo scrimber is one of the most popular engineering bamboo composites, owing to its excellent physical and mechanical properties. In order to investigate the influence of grain direction on the compression properties and failure mechanism of bamboo scrimber, the longitudinal, radial and tangential directions were selected. The results showed that the compressive load–displacement curves of bamboo scrimber in the longitudinal, tangential and radial directions contained elastic, yield and failure stages. The compressive strength and elastic modulus of the bamboo scrimber in the longitudinal direction were greater than those in the radial and tangential directions, and there were no significant differences between the radial and tangential specimens. The micro-fracture morphology shows that the parenchyma cells underwent brittle shear failure in all three directions, while the fiber failure of the longitudinal compressive specimens consisted of ductile fracture, and the tangential and radial compressive specimens exhibited brittle fracture. This is one of the reasons that the deformation of the specimens under longitudinal compression was greater than those under tangential and radial compression. The main failure mode of bamboo scrimber under longitudinal and radial compression was shear failure, and the main failure mode under tangential compression was interlayer separation failure. The reason for this difference was that during longitudinal and radial compression, the maximum strain occurred at the diagonal of the specimen, while during tangential compression, the maximum strain occurred at the bonding interface. This study can provide benefits for the rational design and safe application of bamboo scrimber in practical engineering.

## 1. Introduction

Bamboo is an environmentally friendly biological material with superior physical and mechanical properties, which is widely distributed in Asia, America and other places [1,2]. Bamboo has a short growth cycle and can be harvested every 3–5 years. It has been used in the construction field for a long time. However, due to its small diameter and hollow structure, the further application of bamboo in the construction field is limited [3]. Therefore, some bamboo engineering products, such as bamboo scrimber [4], bamboo laminated timber [5] and bamboo particleboard [6], were explored. Among these, bamboo scrimber has become one of the most well-known bamboo engineering products, due to its high raw material utilization rate (above 90%) and good mechanical properties, and it has been widely used in the construction field [7].

When bamboo scrimber is used as a compression component in a building, its compression properties are important to evaluate its reliability and safety [8]. According to the direction of the bamboo bundle, the bamboo scrimber can be divided into a longitudinal direction (L) and a transverse direction. The direction parallel to the length of the bamboo bundle is the longitudinal direction, and the direction perpendicular to the length of the bamboo bundle is the transverse direction. The transverse direction can be divided into the radial direction (R) and the tangential direction (T), according to the assembly direction of the bamboo bundles, in which the height direction of assembly is the radial direction and the width direction of assembly is the tangential direction, as shown in Figure 1e. Previous studies showed that the compressive properties of bamboo scrimber were anisotropic [9], such that the stress–strain curves of the bamboo scrimber could be divided into elasticity, yield and softening stages under longitudinal compression, while that of bamboo scrimber under tangential compression had only the following two stages: elasticity and yield [10]. The compressive strength of bamboo scrimber under longitudinal compression, meanwhile, was greater than that compressed in a tangential direction (102.1 MPa vs. 54.4 MPa) [9].

Previous studies have mainly focused on the macro-mechanical properties of bamboo scrimber under longitudinal and tangential compression directions. However, there are few studies on the radial compression performance of bamboo scrimber. Furthermore, the failure mechanism of bamboo scrimber under different compression directions has not been systematically studied.

Generally speaking, methods used to study the failure mechanism of materials include failure morphology analysis, stress state analysis, surface strain analysis and numerical methods. Stress state analysis can provide an explanation for the macro-failure mode of materials, while surface strain analysis can visualize the stress concentration phenomenon of materials more intuitively, so as to predict the failure location [11]. Micro-failure morphology can be used to speculate the failure behavior of materials [12], judge the failure mode of materials, and even reveal some details that cannot be observed at the macro scale [13]. Combining the results of stress state analysis with surface strain analysis explains the failure mechanism of a material. The numerical method consists of reproducing the force process of a material using mathematical models, which can directly reflect the stress state and failure process of materials. It is helpful to understand the macroscopic failure mode and predict the location of a failure [14].

In the past, the strain on the surface of materials was difficult to observe. With the application of the digital image correlation (DIC) method, the strain change of a material during the force loading process can be monitored without contact. The DIC method can convert the full-field strain information into an image, which intuitively reflects the stress concentration state of the material, so as to predict the damage location [15] and reveal the cause of the damage [16]. At present, the DIC method has been widely used to study the failure mechanism of wood [17] and bamboo [18].

In this paper, the compressive properties of bamboo scrimber in three directions (longitudinal direction, radial direction and tangential direction) were studied using a mechanical testing machine. During the compression process, the changes in strain field on the surface of the bamboo scrimber were observed by the DIC method. Combined with fracture morphology analysis of bamboo scrimber at macro and micro scales, the failure mechanism of bamboo scrimber under different directions was summarized. The results in this paper can provide a reference for the safe and rational application of bamboo scrimber in the construction field.

## 2. Materials and Methods

### 2.1. Materials

Bamboo scrimber was prepared from moso bamboo (*Phyllostachys pubescens*), 3–5 years old, harvested from Anhui Province, China. The manufacturing process of bamboo scrimber included the following six steps: splitting, fluffing, heat treatment, impregnating, assembly, and hot pressing. Firstly, the bamboo culms were longitudinally split into two semicircular bamboo tubes with a length of 2.4 m (Figure 1a). Without removing the inner and outer layers, the bamboo tubes were then pushed into a fluffing machine along the longitudinal direction to obtain the bamboo fiber bundles (Figure 1b). The bamboo fiber bundles were heated using steam at 180 °C for 2 h (Figure 1c), and were then immersed into phenol formaldehyde resin (PF, type: PF162510 with 45% of solid content, 36 CPs of viscosity, 10~11 pH, Beijing Dynea chemical industry Co., Ltd., Beijing, China) for 10~15 min at room temperature (Figure 1d), before being dried for 8 h at 45 °C. Finally, the bamboo fiber bundles were assembled along the longitudinal direction (Figure 1e) and hot pressed with a pressure of 4.0 MPa at 140 °C for 2 h (Figure 1f).

The specimens (Figure 2a) were divided into three groups according to grain direction, including longitudinal specimens L1~L54 (Figure 2b), tangential specimens T1~T54 (Figure 2c) and radial specimens R1~R54 (Figure 2d), with 54 specimens in each group. The dimensions of each sample were 20 mm × 20 mm × 30 mm, as shown in Figure 2. Before the test, all specimens were conditioned at 25 °C and 65% RH to arrive at an equilibrium moisture content. After equilibrium, the average air-dry density of bamboo scrimber specimens was 1.15 g/cm^3^.

### 2.2. Test Methods

A universal test machine with a load capacity of 100 KN (Model: 5582, Instron Co., Ltd., Norwood, MA, USA) was used to evaluate the compression properties of each specimen (Figure 3b). The DIC method was used to collect the strain on the specimen surface during compression (Figure 3c). In order to facilitate observation, speckles were drawn on the surface of each bamboo scrimber specimen (Figure 3a). Before speckle drawing, the observation surface was ground with 100 mesh sandpaper, and then sprayed with a layer of white primer less than 0.5 mm thick to reduce the impact of surface roughness on the experimental results. During the test, the loading rate was 1 mm/min, the DIC image acquisition speed was 2 frames per second, the image pixels of each DIC image were 4896 × 3264 pixels, and the displacement accuracy of the specimen surface obtained by DIC was 0.01 pixel. When the load dropped to 70% of the maximum load or obvious cracks appeared on the specimen surface, the experiment was stopped [19].

Two strain gauges (Model: BX120-3AA, Xingtai City, Hebei Province, China) were attached at the middle positions of two opposite sides of the specimen to measure the strain. The date was collected by a multi-channel data acquisition instrument (TDS-530, Tokyo, Japan). The final result was the average of experimental tests of two strain gauges.

The compressive strength (*σ*) and elastic modulus (*E*) of the specimen can be calculated by Equations (1) and (2) [19,20], as follow:(1)$$\sigma =\frac{P_{max}}{bt}$$
(2)$$E=\frac{\Delta P}{bt\Delta \xi }$$
where *σ* is the compressive strength (MPa), *P_max_* is the maximum load (kN), *b* is width of specimens (mm), *t* is the thickness of specimens (mm), *E* is the elastic modulus (GPa), ∆*P* is the load increment at the elastic stage of the stress–strain curve (N), ∆*ζ* is the corresponding strain increment at the mid-span cross section (mm).

### 2.3. Microscopic Fracture Morphology

The micro-fracture morphology of longitudinal, tangential and radial specimens was observed by a scanning electron microscope (SEM, Hitachi S-4800, Tokyo, Japan) with low vacuum and an acceleration voltage of 10 kV, to investigate the compression mechanism of bamboo scrimber.

## 3. Results and Discussion

### 3.1. Compression Performance

Figure 4 shows the load–displacement curves of the bamboo scrimber compressed in the longitudinal direction (L), tangential direction (T) and radial direction (R). The curves of L, T and R all contained the following three stages: elastic stage (I), yield stage (II) and failure stage (III). In the elastic stage, the load increases linearly with the increase in displacement. In the yield stage, the load increases nonlinearly with the increase in displacement. In the failure stage, the load decreases with the increase in displacement. The critical point of the elastic stage and the yield stage is located by the second derivative of the curve, while the critical point of the yield stage and the failure stage is the maximum load points. In addition, the maximum displacements of L, T and R were 2.85 mm, 0.58 mm and 0.67 mm, respectively. The compression deformation of L was larger than those of T and R. However, for natural bamboo, the compression deformation in the radial direction was much larger than that in the longitudinal direction [21]. This difference was due to the changes in the micro-structure and chemical components of the bamboo scrimber [22].

Figure 5 shows the average compressive strength and elastic modulus of bamboo scrimber in three grain directions, with each value being the average of 54 values. Different letters represent significant differences, and the same letters represent no significant differences. The compressive strength of L, T and R was 94.67 MPa, 45.86 MPa and 48.80 MPa, respectively, and the elastic modulus of L, T and R was 14.61 GPa, 2.81 GPa and 3.18 GPa, respectively. The compressive strength of L was 2.06 and 1.94 times that of T and R, respectively, and the elastic modulus of L was 5.20 and 4.59 times that of T and R, respectively. There were no significant differences in compressive strength and elastic modulus between T and R. The reason for this was that the compressive strength and elastic modulus in the longitudinal direction of the bamboo scrimber mainly depended on the characteristics of the bamboo fibers, while that of the bamboo scrimber in the radial and tangential directions was mainly determined by the bonding strength of the adhesive [9].

### 3.2. Macroscopic Failure Morphology

In order to study the failure modes and reveal the failure mechanism of bamboo scrimber in three grain directions, the macroscopic failure morphology of each of the specimens was observed and divided into I~V modes (Figure 6, Figure 7 and Figure 8). The proportion of each failure mode in the different grain direction groups is shown in Table 1.

Figure 6 shows the failure modes of the longitudinal compression specimens. The shape of the crack in mode I looks similar to the letter “Y”. This crack first appeared in the middle of the specimen. As the load increased, the crack expanded along the diagonal direction parallel to the 45° angle, then the middle crack spread downwards, in a pattern similar to the letter “Y”. The number of mode I failure specimens accounts for 50% of the total number. In addition, obvious fiber wrinkles and bulges could be observed on the failure surface. This failure mode is shear failure [11].

The crack in mode II was an oblique line along the diagonal direction, and the angle between the oblique line and the horizontal direction was about 60°. In this failure mode, the crack first appeared at the two diagonal corners of the specimen, and then developed along the diagonal direction. The number of mode II failure specimens accounts for 26% of the 54 specimens. This failure mode is also shear failure [9].

Similarly to mode II, mode III was also an oblique crack, but the angle of the crack in mode III was different from that in mode II. The angle between the oblique crack and the horizontal direction of mode III was about 45°. The number of specimens with this failure mode accounts for 22% of the 54 specimens. This failure mode is also shear failure.

Mode IV was a vertical crack. In this failure mode, the crack propagated along the bonding interface. With the increase in pressure, the sample divided into two parts. This failure mode may be caused by uneven impregnation, assembly and low bonding strength during the processing of bamboo scrimber, and it has the greatest impact on the properties of bamboo scrimber. This failure mode is associated with interlaminar separation failure. In this study, only one specimen was in this failure mode.

In summary, the failure modes I, II and III were shear failure, and failure mode IV was interlayer separation failure. Thus, shear failure was the main failure mode of bamboo scrimber under compression in the longitudinal direction.

Figure 7 shows the failure modes of bamboo scrimber under tangential compression. Similarly to longitudinal compression, bamboo scrimber under tangential compression showed “Y” cracks (mode I), diagonal cracks (mode II), 45° oblique cracks (mode III) and vertical cracks (mode IV). The proportion of each failure mode was 11%, 19%, 15% and 22%, respectively. The number of specimens in mode IV, under tangential compression, was greater than that under longitudinal compression. Differently from longitudinal compression, there was a mixed failure mode V under tangential compression, which consisted of the combination of shear failure and interlayer separation. This failure mode made up the largest proportion of the tangential compression specimens, which was 34%. Thus, interlayer separation failure was the main failure mode of the specimens under compression in the tangential direction. The reason for this was that the bamboo bundles were only connected by gluing in the transverse direction, so the glue layer and the bamboo bundle were easier to separate under tangential compression [23].

Figure 8 shows the failure modes of the bamboo scrimber under radial compression. Similarly to longitudinal compression and tangential compression, the failure mode also showed “Y” cracks (mode I), diagonal cracks (mode II), vertical cracks (mode IV) and mixed cracks (mode V). Among these, the proportion of mode I was the highest, which was 50%, and the proportions of mode II, mode IV and mode V were 10%, 15% and 25%, respectively. It is worth noting that the vertical crack (Figure 8, mode IV) in radial compression failure was different from that in longitudinal compression and tangential compression. This crack did not appear at the bonding interface, but was formed by the intersection of two oblique cracks. Thus, shear failure was the dominant failure mode for specimens that were compressed along the radial direction.

### 3.3. Microscopic Failure Morphology

In order to reveal the reason for the differences in macro-compression properties in three grain directions of bamboo scrimber, the microscopic damage morphologies of the longitudinal, tangential and radial compression specimens were analyzed. A representative sample was selected in each group; they were L17, T2 and R42, respectively (Figure 9).

When compressed in the longitudinal direction, the bamboo fibers were distorted (Figure 9(a2)), and a few bamboo fibers showed characteristics of shear failure and fiber debonding (Figure 9(a2)). It can be judged that the failure of the bamboo fibers during longitudinal compression was ductile [24]. However, when compressed in the tangential and radial directions, the fibers were broken neatly and without deformation (Figure 9(b2,c2)), which means that the failure of the fibers was brittle. This is one of the reasons why the deformation of specimens during longitudinal compression was greater than that of the specimens under tangential and radial compression (Figure 4).

On the other hand, the parenchyma cells showed brittle shear failure in all three groups (Figure 9(a3,b2,c3)). This was correlated with their short and thin-walled structures [25]. In addition, as shown in Figure 9(b3), the micro-failure morphology of the specimens at Position 2 was smooth and without fiber breakage, which further confirmed that interlayer separation failure occurred here.

### 3.4. Failure Mechanism

#### 3.4.1. Force Analysis

Through the analysis of the macroscopic and microscopic failure morphology of the bamboo scrimber under different grain directions, it was found that shear failure was the main failure mode. This was related to the stress distribution inside the specimen during the compression process. As shown in Figure 10, the stress (*p**_α_*) in any section of the specimen could be decomposed into normal stress(*σ_α_*) and shear stress(*τ_α_*). Shear stress dominates the compressive failure of bamboo scrimber [26]. According to the law of force composition, the stress *p_α_* was as follows:(3)$$p_{a}=\sigma_{0}\cos\alpha$$

The shear stress was [11] as follows:(4)$$\tau_{\alpha }=\sigma_{0}\sin\alpha \cos\alpha$$
where *α* is the angle between the loading force and the normal direction of the section; *F* is the compression load; *σ*_0_ is the normal stress of the cross section.

It can be observed from Equation (3) that when the *α* is 45°, the *τ*_α_ is the maximum. Thus, the crack expanded in the direction of the maximum force (*α* = 45°) [27]. However, in the actual compression process, the specimen was easily twisted, leading to deflection of the crack angle, as with mode II (Figure 6).

#### 3.4.2. Strain Field Distribution

According to the above analysis, the main failure mode of the bamboo scrimber under longitudinal compression and radial compression was shear failure, while the main failure mode under tangential compression was layer separation failure. In order to further analyze the reason for this difference, the DIC method was used to monitor the surface strain field of the specimens. During the compression process, the crack morphology can only be observed on a specific surface, which is the surface shown in Figure 6, Figure 7 and Figure 8. In order to analyze the failure mode and surface strain field together, this surface was selected for strain distribution observation. The strain distribution of the observation surface is the same as that of the opposite surface, which is different from the adjacent surface. L17 and T5 were selected as representative samples for shear failure and layer separation failure analysis, respectively.

Figure 11 shows the strain field distribution of the shear failure specimen L17 under the maximum load; its failure mode was mode II with diagonal cracks. Exx represents the strain in the X direction, eyy represents the strain in the Y direction, and exy represents the shear strain. When the load reached the maximum value, the maximum values of exx, eyy and exy all occurred at the diagonal of the specimen, indicating that a significant stress concentration occurred here, which is the location of the initial crack. This result is consistent with the above analysis.

There was a significant difference in the strain field of the tangential compression specimen T5. It can be observed from Figure 12 that the maximum exx, eyy and exy all appeared in the upper left corner of the specimens, and that the initial crack occurred here. On the other hand, the strain distributions of exx and eyy were stratified and parallel to the bonding layer of the specimen, resulting in interlayer separation failure at the bonding interphase. This indicates that the bonding interface affected the stress distribution and was prone to produce stress concentration, resulting in failure [19].

## 4. Conclusions

In order to reveal the influence of grain direction on the compression properties and failure mechanisms of bamboo scrimber, the longitudinal, radial and tangential directions were selected. Moreover, the load–displacement curves, compression properties, macro and micro-failure morphology, stress state and strain field were analyzed. The main conclusions are summarized as follows:(1)The compressive load–displacement curves of bamboo scrimber in longitudinal, tangential and radial directions all contained an elastic, yield and failure stage.(2)The compressive strength of bamboo scrimber along the longitudinal direction was 94.67 MPa, which was 2.06 times that of the tangential specimens and 1.94 times that of the radial specimens. The elastic modulus of the longitudinal specimens was 14.61 GPa, which was 5.20 times that of the tangential specimens and 4.59 times that of the radial specimens.(3)The micro-failure morphology shows that the parenchyma cells showed brittle shear failure in all three directions, and the fiber failure of the longitudinal compressive specimens was ductile fracture, while that of the tangential and radial compressive specimens was brittle fracture. This is one of the reasons that the deformation of the specimens under longitudinal compression was greater than in the tangential and radial directions.(4)Under the compression of three grain directions, the macro-failure modes of bamboo scrimber include five modes, which are the “Y” crack (mode I), diagonal crack at 60° (mode II), diagonal crack at 45° (mode III), vertical crack (mode IV) and mixed crack (mode V). Among these, the longitudinal specimens include the modes I, II, III, and IV, the tangential specimens include the modes I, II, III, IV, and V, and the radial specimens include the modes I, II, IV, and V. The main failure mode of the longitudinal and radial compressive specimens was shear failure, while that of the tangential compressive specimens was interlayer separation failure. The reason for the difference was that the maximum strain occurred at the diagonal of the specimen during longitudinal and radial compression, while the maximum strain occurred at the bonding interface under tangential compression.(5)Bamboo scrimber is a new kind of structural engineering material with broad application prospects. There has been little research on the bonding interface performance, long-term performance and seismic performance of bamboo scrimber when used as a structural material, something to which more attention needs to be paid.

## Figures and Tables

**Figure 1 polymers-13-04223-f001:**
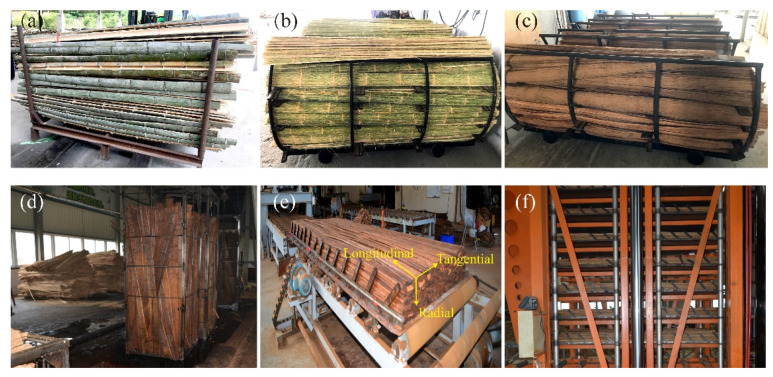
**The** manufacturing process of bamboo scrimber: (**a**) splitting; (**b**) fluffing; (**c**) heat treatment; (**d**) impregnating; (**e**) assembly; (**f**) hot pressing.

**Figure 2 polymers-13-04223-f002:**
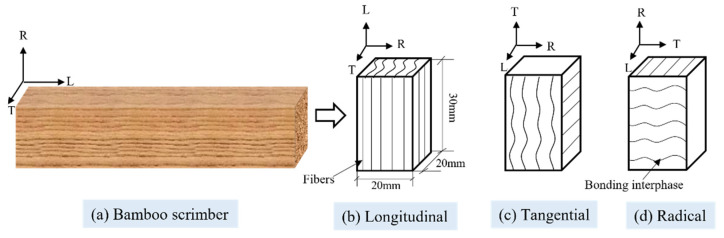
Schematic diagram of specimen preparation.

**Figure 3 polymers-13-04223-f003:**
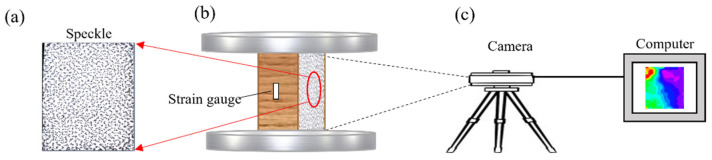
Schematic drawing of compression testing: (**a**) speckle on the specimen surface; (**b**) test schematic; (**c**) DIC.

**Figure 4 polymers-13-04223-f004:**
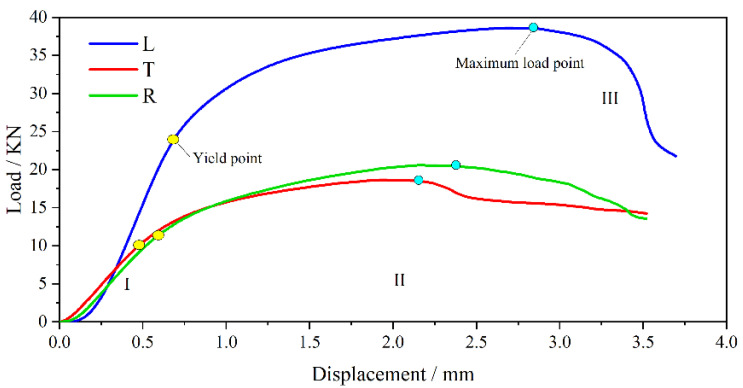
Load–displacement curves of bamboo scrimber in different grain directions. I elastic stage; II yield stage; III failure stage.

**Figure 5 polymers-13-04223-f005:**
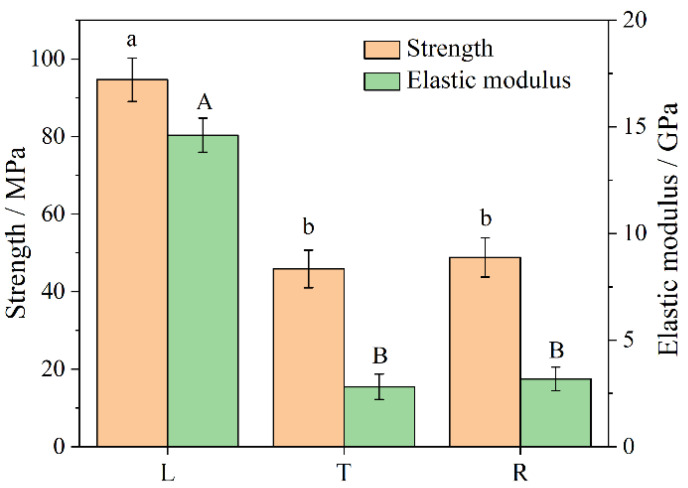
Compressive strength and elastic modulus of bamboo scrimber in three grain directions. Note: Lowercase letters indicate the difference in compressive strength, and uppercase letters indicate the difference in compressive elastic modulus. The same letter indicates insignificant difference (*p* > 0.05), and different letters indicate significant difference (*p* < 0.05).

**Figure 6 polymers-13-04223-f006:**
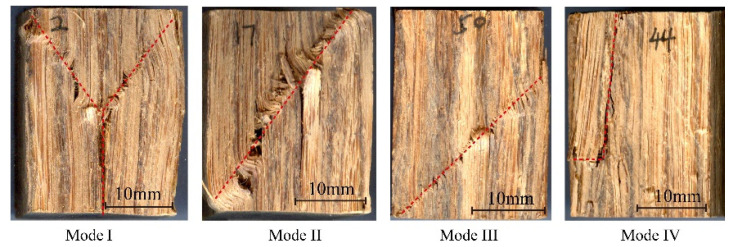
Failure modes of longitudinal compression specimens.

**Figure 7 polymers-13-04223-f007:**
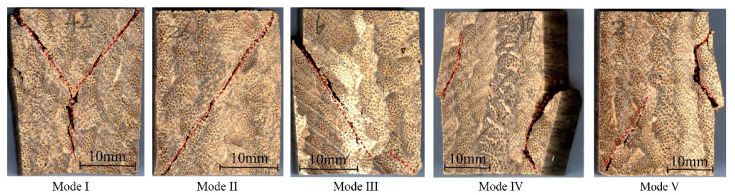
Failure modes of tangential compression specimens.

**Figure 8 polymers-13-04223-f008:**
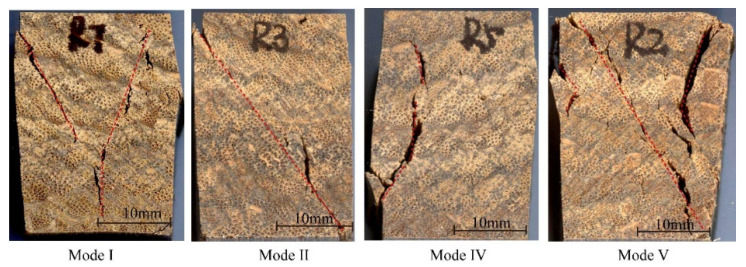
Failure modes of radial compression specimens.

**Figure 9 polymers-13-04223-f009:**
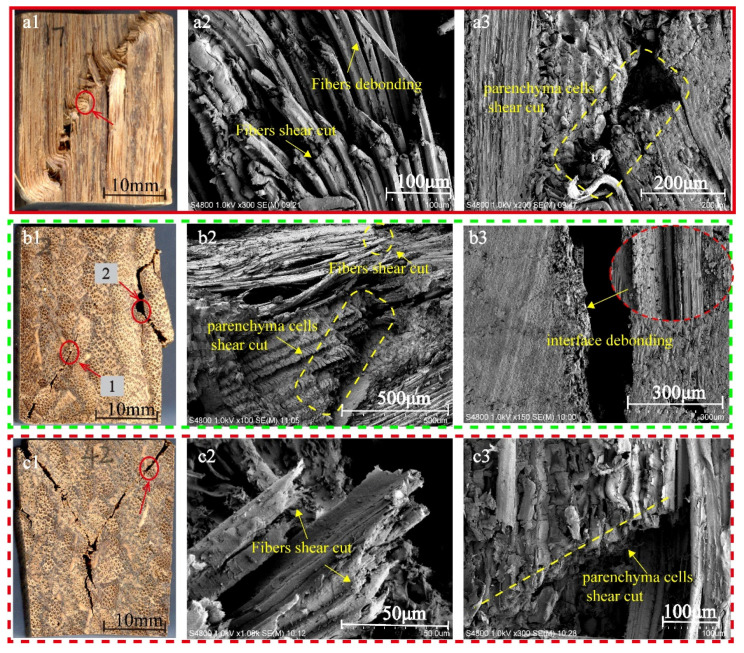
Micro-fracture morphology of bamboo scrimber in three grain directions: (**a1**) sampling location of longitudinal compression specimen; (**a2**) micro-fracture morphology of fibers; (**a3**) micro-fracture morphology of parenchyma cells; (**b1**) sampling location of tangential compression specimen; (**b2**) micro-fracture morphology of fibers and parenchyma cells of position 1; (**b3**) micro-fracture morphology of position 2; (**c1**) sampling location of radial compression specimen; (**c2**) micro-fracture morphology of fibers; (**c3**) micro-fracture morphology of parenchyma cells.

**Figure 10 polymers-13-04223-f010:**
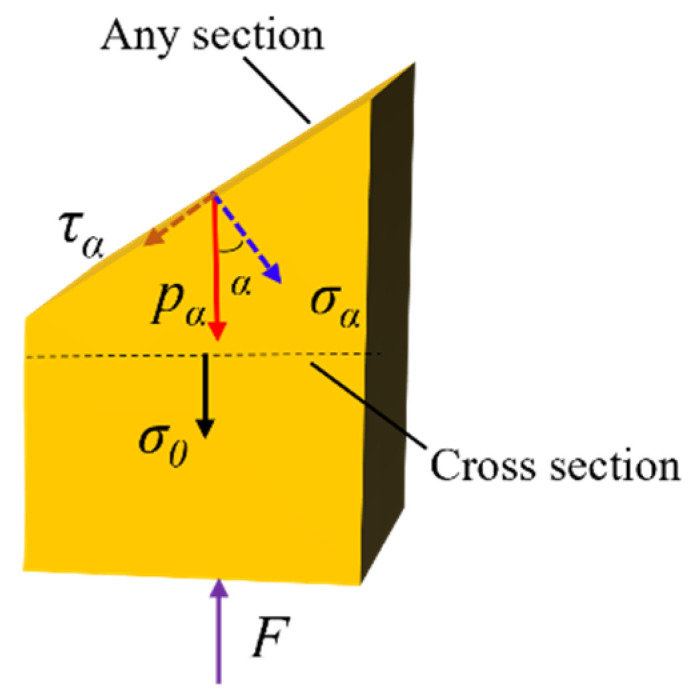
Schematic diagram of stress state during compression.

**Figure 11 polymers-13-04223-f011:**
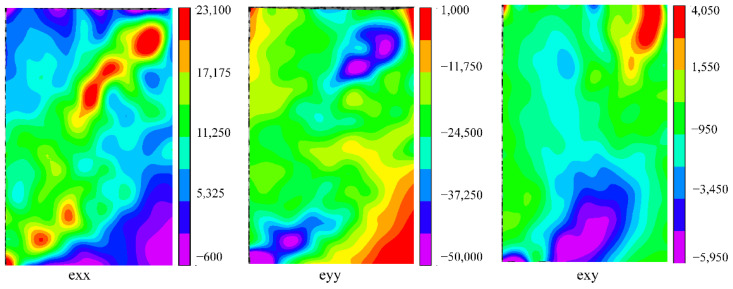
Strain field distribution of specimen L17 under the maximum load (shear failure).

**Figure 12 polymers-13-04223-f012:**
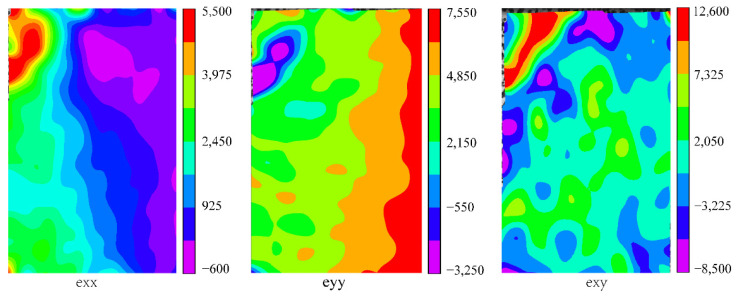
Strain field distribution of specimen T5 under the maximum load (layer separation failure).

**Table 1 polymers-13-04223-t001:** Proportion of failure modes for each group.

Specimen Group	Mode I	Mode II	Mode III	Mode IV	Mode V	Total
L	50%	26%	22%	2%	-	54
T	11%	18%	15%	22%	34%	54
R	50%	9%	-	15%	26%	54

## Data Availability

The date presented in this study are available on request from the corresponding author.

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
