# Peer review of "Compressive Failure Mechanism of Structural Bamboo Scrimber"

_polymers, 2021, doi:10.3390/polym13234223_

Round 1

Reviewer 1 Report

This paper studied the Compressive Failure Mechanism of Structural Bamboo Scrimber. Overall, the research is conducted well, I got some comments which I believe will improve the presentation of the research results:

In addition to the general description in lines 56-64, please include some specific references studying the shear failure/damage of composites using stress state analysis by numerical methods, such as doi: 10.1016/j.ceramint.2015.09.085

The stress decomposition in Fig. 10 indicates τα = pα sin?. Not sure how did you get τα = pα sin?cos??

‘Red indicates a positive strain value, and blue indicates a negative strain value. A darker color indicates an increase in the strain value’. This piece of information is not necessary. It should be easily read through the figure legend. The width of the legend in strain contours (Figs. 11-12) can be enlarged, to better show the colours vs values.

Reviewer 2 Report

The manuscript focus on the compressive failure mechanism of structural bamboo scrimbers. The manuscript is well written. I have some minor comments before further processing:

  • Please do not use citation pockets (e.g. [4-6]) but rather cite each reference individually (e.g. as described in [5]). If it is not possible please delete redundant references,
  • I would like to see the scale bars on fig. 6-9 to see the dimensions of failures,
  • I suggest to add some perspectives for future studies in conclusion section.

Reviewer 3 Report

The manuscript presents a study on the mechanical behavior of structural bamboo scrimber. A special focus is put on the compressive failure mechanisms of bamboo scrimber where the loading is along three different directions (i.e., longitudinal, radial, and tangential). The macroscopic failure morphology is summarized and analyzed with the aid of the strain field obtained by DIC. Different failure mechanisms are revealed and related to the micro- and macro-observations.

Overall speaking, the structure of the paper is clear and the written is acceptable for the readers to understand the whole study. These experimental results, as well as the analysis of failure mechanisms, can provide benefits for the rational design and safe application of bamboo scrimber in practical engineering, as the authors claimed in the Abstract. Therefore, the reviewer can recommend it for publication after the following comments are dealt with.

  1. For the load-displacement curves, how do the authors define and identify the elastic stage, yield stage and failure stage? Is there some plasticity in the yield stage? In fact, the curves of T and R also have a decrease phase which may be considered as a so-called failure stage.

  1. When can the macro-crack be identified during the loading process? The strain field is measured by DIC when the load is maximum. The reviewer wants to know if in this point the macro-cracks have appeared.
  2. What do the symbols “a, A, b, B” mean in Figure 5? Do the measurements of strength and elastic modulus consider the change of cross section due to longitudinal compression?
  3. Can the authors give more explanation on the difference between the tangential and radical directions? It is not easy to understand it directly from Figure 2. In the reviewer’s opinion, they should have the same mechanical properties.
  4. Figure 10 is difficult to be understood. What does the solid curve mean? Why does the curve have this type? More explanation is needed here.
  5. In the figures of strain field distribution, what does the value mean? It seems that the color bar does not represent the magnitude of strain. Does the strain map shown in these figures correspond to the whole side face? Can the author provide more information on the technical details of DIC, especially the precision? Does the surface roughness influence the DIC results? The surface roughness seems different for the three different directions.

Round 2

Reviewer 1 Report

Thank you for the response and revision, just few minor issuses: Line 69, grammar mistake 'methods was';  Line 405, Ref. 14, author surname and given name, volume (issue) and page number, doi are incorrect.

Author Response

I'm sorry for my mistakes. The grammatical errors and reference errors mentioned have been revised in the manuscript, see line 69 and line 405.